# Contribution to the Knowledge of Rocky Plant Communities of the Southwest Iberian Peninsula

**DOI:** 10.3390/plants10081590

**Published:** 2021-08-02

**Authors:** Ricardo Quinto Canas, Ana Cano-Ortiz, Giovanni Spampinato, Sara del Río, Mauro Raposo, José Carlos Piñar Fuentes, Carlos Pinto Gomes

**Affiliations:** 1Faculty of Sciences and Technology, University of Algarve, Campus de Gambelas, 8005-139 Faro, Portugal; 2CCMAR—Centre of Marine Sciences (CCMAR), University of Algarve, Campus de Gambelas, 8005-139 Faro, Portugal; 3Department of Animal and Plant Biology and Ecology, Section of Botany, University of Jaén, Las Lagunillas s/n, 23071 Jaén, Spain; anacanor@hotmail.com (A.C.-O.); jcpfuentes@gmail.com (J.C.P.F.); 4Department of AGRARIA, “Mediterranea” University of Reggio Calabria, Località Feo di Vito, 89122 Reggio Calabria, Italy; gspampinato@unirc.it; 5Department of Biodiversity and Environmental Management (Area of Botany), Mountain Livestock Farming Institute (Joint Center CSIC-ULE), Faculty of Biological and Environmental Sciences, Campus of Vegazana, University of León, s/n, 24071 León, Spain; sriog@unileon.es; 6Department of Landscape, Environment and Planning, Mediterranean Institute for Agriculture, Environment and Development (MED), School of Science and Technology, University of Évora, Rua Romão Ramalho, n° 59, 7000-671 Évora, Portugal; mraposo@uevora.pt (M.R.); cpgomes@uevora.pt (C.P.G.)

**Keywords:** phytosociology, numerical analysis, *Phagnalo saxatilis-Rumicetea indurate* class, southern Portugal, *Dianthus crassipes*, *Antirrhinum onubensis*

## Abstract

The rocky habitats of southern Portugal are ecosystems with extreme xericity conditions, associated with special abiotic strains. In these unstable ecological conditions, a considerable diversity of plant communities occurs. The objective of this study, carried out in the Algarve and Monchique, and the Mariánica Range biogeographical sectors, is to compare chasmo-chomophytic communities of the southwestern Iberian Peninsula, using a phytosociological approach (Braun–Blanquet methodology) and numerical analysis (hierarchical cluster analysis). From these results, two new communities were identified, *Sanguisorbo rupicolae-Dianthetum crassipedis* and *Antirrhinetum onubensis*, as a result of floristic and biogeographical differences from other associations already described within the alliances *Rumici indurati-Dianthion lusitani* and *Calendulo lusitanicae-Antirrhinion linkiani*, both included in the *Phagnalo saxatilis-Rumicetea indurate* class.

## 1. Introduction

Habitats are very important natural or semi-natural places that need to be continuously studied to preserve them and their inhabitants, especially the plant endemic species that characterize them [1,2]. In fact, different habitats in the world are threatened by different types of pressures and threats [3,4,5,6,7,8,9,10]. Among the wide spectrum of habitats, we focus on plant communities that develop on rocky habitats, which are often associated with crests, cliffs, lithosols, rocky soils or rock outcrops. In these edaphoxerophilous biotopes that appear to be evidence of extreme xericity conditions, namely great edaphic drought as a result of the soil’s reduced capacity for water retention [5,11]. In fact, this environment offers the ideal conditions for a highly specialized type of vegetation: the chasmo-chomophytic communities, which encompass rupicolous vegetation that colonizes earthy broad crevices, rocky soils and lithosols (chomophytes) or narrow rocky fissures (chasmophytes).

The perennial chasmo-chomophytic communities of the southwestern Iberian Peninsula are included in the *Phagnalo saxatilis-Rumicetea indurati* vegetation class [12,13]. The *Phagnalo saxatilis-Rumicetalia indurati* is the only order of this class and is further divided into six alliances in the Iberian Peninsula [13]. Because of the substrata nature given to the descriptions of the chasmo-chomophytic communities within the *Phagnalo saxatilis-Rumicetalia indurati* order, such as schist or greywackes, limestone and dolomite, Costa et al. [12] recognized the presence of two alliances in southern Portugal: *Rumici indurati-Dianthion lusitani*, which is ascribed to the communities developing on fissures of siliceous rocks of the West Iberian Mediterranean and Oroiberian territories, whereas the communities growing in the Dividing Portuguese Sector and Arrábida range (central and western Portugal), typically associated with limestone and dolomitic substrates, are classified within the *Calendulo lusitanicae-Antirrhinion linkiani* class.

The present paper aims to provide new knowledge of the chasmo-chomophytic communities of the southwestern Iberian Peninsula, included in the *Phagnalo saxatilis-Rumicetea indurati* class. The phytosociological and syntaxonomical vegetation analysis of the chasmo-chomophytic communities dominated by *Dianthus crassipes* (*Rumici indurati-Dianthion lusitani* alliance) and *Antirrhinum onubensis* (*Calendulo lusitanicae-Antirrhinion linkiani* alliance) allow us to distinguish two new communities: *Sanguisorbo rupicolae-Dianthetum crassipedis* and *Antirrhinetum onubensis.*

## 2. Materials and Methods

### 2.1. Study Area

Located in the southwest of the Iberian Peninsula, the study area covers two distinct units: the first, in the southern part of the Algarve and Monchique biogeographical sector, encompasses the sub-littoral lower altitude reliefs of Algarve limestone—Barrocal algarvio (maximum 480 m high), where there is frequent presence of basic rocky outcrops. The second includes the schist, greywacke and quartzite cliffs of the low-altitude range (with an altitude of less than 400 m) of the lower Guadiana valley, in the southwestern part of the Mariánica Range Sector. According to the most recent study of Peninsula Iberica bioclimatic characterization by Rivas-Martínez et al. [14], the study area is classified as mediterranean pluviseasonal oceanic, dry to subhumid thermomediterranean bioclimate.

### 2.2. Data Collection

Field research was carried out from 2011 to 2020. Phytosociological relevés were collected according to the Zurich–Montpellier phytosociological method [12,13,15,16,17], where we found two new distinct associations in the Algarve and Monchique, and Mariánica Range biogeographical sectors (Figure 1), based on the comparison of the phytosociological relevés, performed in Table 1. Following Biondi [18], each relevé is a floristically and ecologically homogeneous plant community that represents the plant association on the ground. Within this definition, for each relevé, all plants that are found in an area whose floristic, structural and ecological conditions are homogeneous, were identified and assigned a quantitative value or index for their coverage, using the conventional abundance–dominance scale of Braun–Blanquet.

### 2.3. Nomenclature

Syntaxonomical typologies followed Rivas-Martínez [13,26,27], Costa et al. [12] and Mucina et al. [28]. Plant identification follows Coutinho [29], Franco [30], Franco and Rocha Afonso [31], Castroviejo [32] and Valdés et al. [33]. Taxonomic nomenclature was updated using Iberian lists elaborated by Rivas-Martínez et al. [27], Sequeira et al. [34] and Costa et al. [12]. The biogeographical and bioclimatological information was collected according to Rivas-Martínez et al. [14,19], and substratum affinity information was collected from the literature: Quinto-Canas [35], Meireles [36], Orellana and Galán de Mera [22] and Lopes [24]. The phytosociological name of the new vegetation unit is given according to the International Code of Phytosociological Nomenclature [37].

### 2.4. Data Analysis

For statistical data processing of the samples, we first generated a data matrix that included 64 relevés and 135 species from our field sampling (Table 1 and Table 2; association 1; Figure 2, clusters 1–8; Table 3, association 2; Figure 2, clusters 9–16) and relevés taken from the literature [20,21,22,23,25,35,36]. The matrix was subjected to the unweighted pair-group method using arithmetic averages (UPGMA), with Bray–Curtis distance, to produce the dissimilarity measure, using the software Primer 6 [38,39].

The transformation of Braun–Blanquet‘s abundance–dominance values follows Van der Maarel [40]. This transformation is required as a solution for converting the non-numerical values into numerical scale and in this form used as input data for numerical analysis, with the following equivalence: r = 1, + = 2, 1 = 3, 2 = 4, 3 = 5, 4 = 6 and 5 = 7.

## 3. Results and Discussion

### 3.1. Classification of Southwestern Iberian Peninsula Chasmo-Chomophytic Communities

The dendrogram and the synoptic table reveal a clear separation between all phytosociological chasmo-chomophytic communities of the Southwestern Iberian Peninsula, included in both alliances: *Rumici indurati-Dianthion lusitani* and *Calendulo lusitanicae-Antirrhinion linkiani*. The cluster analysis (Figure 2) produces two main groups of associations (group A and group B), which represent eight community types: *Sanguisorbo rupicolae-Dianthetum crassipedis* (clusters 1–8), *Antirrhinetum onubensis* (9–16), *Phagnalo saxatilis-Rumicetum indurati* (clusters 17–21), *Digitali thapsi-Dianthetum lusitani* (clusters 22–31), *Silene acutifoliae-Dianthetum lusitani* (clusters 32–40), *Sileno montistellensis-Dianthetum lusitani* (clusters 41–44), *Phagnalo saxatilis-Dianthetum barbati* (clusters 45–51) and *Sileno longiciliae-Antirrhinetum linkiani* (clusters 52–64).

Group A has a high dissimilarity in relation to the other associations and includes relevés dominated by *Dianthus lusitanus*. The *Digitali thapsi-Dianthetum lusitani* is an association co-dominated by *Dianthus lusitanus* and *Digitalis thapsi* which occur in the mesomediterranean to supramediterranean bioclimatic stages of the Lusitania and Extremadura, Carpetana and León, and Oroiberian Subprovinces, on rocky fissures of schist, quartzite and granite [21]. The *Silene acutifoliae-Dianthetum lusitani* occur mostly on quartzitic outcrops in the mesomediterranean belt of the São Mamede mountains (São Mamede Sierran District, Oretana Range and Tajo Sector), well characterized by the presence of *Silene acutifolia* [22]. The *Sileno montistellensis-Dianthetum lusitani* is found in submediterranean areas, with the supramediterranean to oromediterranean themotype, on granitic fissures of the Estrela mountain biogeographical territory, and is characterized by the presence of *Silene x montistellensis* (hybrid between *Silene acutifolia* and *Silene foetida*) [36].

The cluster analysis also shows a group of relevés clearly separated from the rest, which are included in the cluster group B, divided into two subgroups. The relevé cluster subgroup B1 corresponds to the association *Antirrhinetum onubensis*, which is proposed here as a new association, largely confined to the limestones of Algarve District (Algarve and Monchique Sector). The subgroup B2 comprises samples ascribed to both silicicolous and calcicolous associations included in the alliances *Rumici indurati-Dianthion lusitani* (such as *Phagnalo saxatilis-Rumicetum indurati* and *Sanguisorbo rupicolae-Dianthetum crassipedis*) and *Calendulo lusitanicae-Antirrhinion linkiani* (such as *Phagnalo saxatilis-Dianthetum barbati* and *Sileno longiciliae-Antirrhinetum linkiani*), respectively. The floristic similarities between these four associations are a result of the high presence of *Phagnalon saxatile* and *Sanguisorba rupicola* in its characteristic species set, both with indifferent soil preferences, and also, in the scope of companion species, the presence of *Dactylis hispanica* subsp. *lusitanica*, *Sedum forsterianum*, *Hyparrhenia sinaica*, *Helichrysum stoecha*, *Sedum sediforme*, *Melica minuta*, *Polypodium cambricum* and *Sedum album*. The *Phagnalo saxatilis-Rumicetum indurati* is dominated by *Phagnalon saxatile* and *Rumex induratus*, and is widely distributed throughout the thermomediterranean to supramediterranean areas of the West Iberian Mediterranean, and Coastal Lusitania and West Andalusia Provinces. The *Sanguisorbo rupicolae-Dianthetum crassipedis*, a new association physiognomically characterized by *Dianthus crassipes*, occurs in the thermomediterranean to mesomediterranean dry areas of the East Mariánica District (Mariánica Range Sector). The *Phagnalo saxatilis-Dianthetum barbati*, a calcicolous community characterized by the co-dominance of *Dianthus barbatus* and *Phagnalon saxatile*, is found in the northern part of the Divisório Portuguese Sector [24]. The *Sileno longiciliae-Antirrhinetum linkiani*, which has been described by Ladero et al. [25] in the limestones of the Divisório Portuguese Sector and Arrabida Sierran District (Ribatejo and Sado Sector), is characterized by the dominance of species from the *Calendulo lusitanicae-Antirrhinion linkiani* alliance, such as *Antirrhinum linkianum*, *Silene longicilia*, *Calendula suffruticosa* subsp. *lusitanica*, *Biscutella valentina*, *Arabis sadina*, *Avenula lodunensis* subsp. *occidentalis* and *Saxifraga cintrana*.

As evidenced in Table 1, in the main characteristics and companion species group, the associations of subgroup B2 encompass differential and territorial species, which support the ecological concept of divisions proposed by Ladero et al. [25] for the *Phagnalo saxatilis-Rumicetalia indurati* order in the West Iberian Mediterranean territories: the alliance *Rumici indurati-Dianthion lusitani* for the siliceous rocks and *Calendulo lusitanicae-Antirrhinion linkiani* for the outcrops of limestone and dolomitic rocks.

### 3.2. Description of the New Chasmo-Chomophytic Associations

**I—*****Sanguisorbo rupicolae-Dianthetum crassipedis*****ass. nova hoc. loco** (Table 2; clusters 1–8).

The relevés of the new association *Sanguisorbo rupicolae-Dianthetum crassipedis* (holotypus Table 2, relevé 8) appear to be clearly defined in group A (clusters 1–8; Figure 2). It is a perennial chasmo-chomophytic community, which develops on acid rocky fissures of schist or greywackes and quartzitic outcrops of the Guadiana basin, in the southeastern part of Portugal (East Mariánica District, Mariánica Range Sector). The *Sanguisorbo rupicolae-Dianthetum crassipedis* is an association characterized by *Dianthus crassipes* and *Sanguisorba rupicola* [41] (Figure 3). As shown in Table 2, the floristic composition also contains other chasmo-chomophytic species from the *Phagnalo saxatilis-Rumicetea indurati* class, such as *Phagnalon saxatile* and *Rumex induratus*. The rupicolous character is emphasized by the presence, in the companion species group, of chasmophytic elements from the *Asplenietea trichomanis* class, such as *Cheilanthes maderensis*, *Cheilanthes guanchica* and *Cosentinia vellea*. Regarding the xerophilous position, this new association develops on the most exposed sector of the cliffs or crests and is distributed in holm oak woodland domains of the *Myrto communis-Quercetum rotundifoliae* or, in the driest siliceous areas of the lower part of the Guadiana valley, on potential areas of the sclerophyllous shrubs of *Juniperus turbinata*, from the *Phlomido purpureae-Juniperetum turbinatae*.

We place this new *Dianthus crassipes* community, at association rank, within the *Rumici indurati-Dianthion lusitani* alliance, which comprises the heliophilous and xerophilous chasmo-chomophytic communities growing on acid siliceous rocks of the West Iberian Mediterranean Province [12].

**II—*Antirrhinetum onubensis* ass. nova hoc. loco** (Table 3; clusters 9–16).

The new association *Antirrhinetum onubensis* (holotypus Table 3, relevé 6) occurs in the thermomediterranean, dry to sub-humid belts of the Algarve District (Algarve and Monchique Sector), on sub-coastal cliffs or rocky fissures of limestones from southern Portugal (Barrocal algarvio reliefs) (Figure 4). Thus, *Antirrhinetum onubensis* is a calcicolous association developed in potential areas of the edaphoxerophilous woodlands dominated by *Quercus rotundifolia* (*Rhamno oleoidis-Quercetum rotundifoliae*), *Juniperus turbinata* (*Aristolochio baeticae-Juniperetum turbinatae*) and *Ceratonia siliqua* (*Vinco difformis-Ceratonietum siliquae*).

The new *Antirrhinum onubensis* chasmo-chomophytic community proposed here is characterized by other species from the class *Phagnalo saxatilis-Rumicetea indurati*, such as *Sedum mucizonia* and *Phagnalon saxatile*, and other thermophile differential species (Table 1): *Asplenium petrarchae*, *Genista hirsuta* subsp. *algarbiensis*, *Aristolochia baetica*, *Elaeoselinum tenuifolium*.

We place the *Antirrhinetum onubensis* at the association rank, in the *Calendulo lusitanicae-Antirrhinion linkiani* alliance, which encompasses the rupicolous vegetation of calcareous crevices of central and western Portugal (in the Divisório Portuguese Sector and Arrabida Sierran District) [10,26]. Nevertheless, according to our results, we propose to modify the alliance diagnosis, extending over the biogeographical territories of the southern Portugal (Algarve District, Algarve and Monchique Sector). Moreover, this community should be classified as a priority habitat from Habitats Directive 92/43/EEC, under the Natura 2000 code: *6110 Rupicolous calcareous or basophilic grasslands of the *Alysso-Sedion albi*, from the Annex I habitat types of the Council Directive 92/43/EEC of 21 May 1992.

### 3.3. Syntaxonomical Scheme

*PHAGNALO SAXATILIS-RUMICETEA INDURATI* (Rivas Goday & Esteve 1972) Rivas-Martínez, Izco & Costa 1973
*PHAGNALO SAXATILIS-RUMICETALIA INDURATI* Rivas Goday & Esteve 1972
*Rumici indurati-Dianthion lusitani* Rivas-Martínez, Izco & Costa ex V. Fuente 1986
*Phagnalo saxatilis-Rumicetum indurati* Rivas-Martínez ex F. Navarro & C. Valle in Ruiz 1986*Digitali thapsi-Dianthetum lusitani* Rivas-Martínez ex V. Fuente 1986*Silene acutifoliae-Dianthetum lusitani* Vicente & Galán 2008*Sileno montistellensis-Dianthetum lusitani* Rivas-Martínez 1981 corr. Ladero, Rivas-Martínez, Amor, M.T. Santos & Alonso 1999*Sanguisorbo rupicolae-Dianthetum crassipedis* Quinto-Canas, Cano-Ortiz, Spampinato, del Río, M. Raposo, Piñar Fuentes & Pinto-Gomes ass. nova hoc loco*Calendulo lusitanicae-Antirrhinion linkiani* Ladero, C. Valle, M.T. Santos, Amor, Espírito Santo, Lousã & J.C. Costa 1991
*Phagnalo saxatilis-Dianthetum barbati* C. Lopes, Pinto-Gomes, Lousã & Ladero 2012*Sileno longiciliae-Antirrhinetum linkiani* Ladero, C. Valle, M.T. Santos, Amor, Espírito Santo, Lousã & J.C. Costa 1991*Antirrhinetum onubensis* Quinto-Canas, Cano-Ortiz, Spampinato, del Río, M. Raposo, Piñar Fuentes & Pinto-Gomes ass. nova hoc loco

## Figures and Tables

**Figure 1 plants-10-01590-f001:**
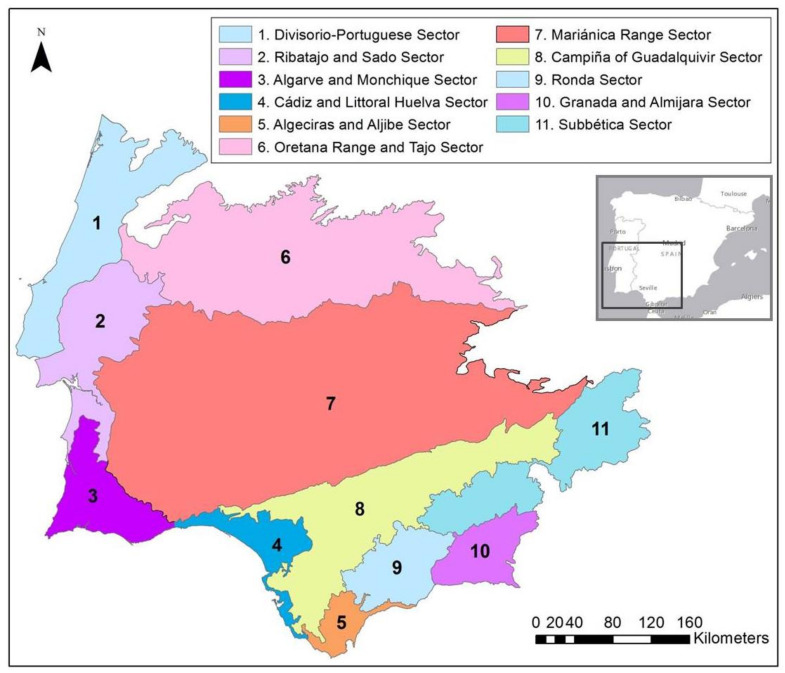
Biogeographical map of the southwest of the Iberian Peninsula at sector level, following Rivas-Martínez et al. [19].

**Figure 2 plants-10-01590-f002:**
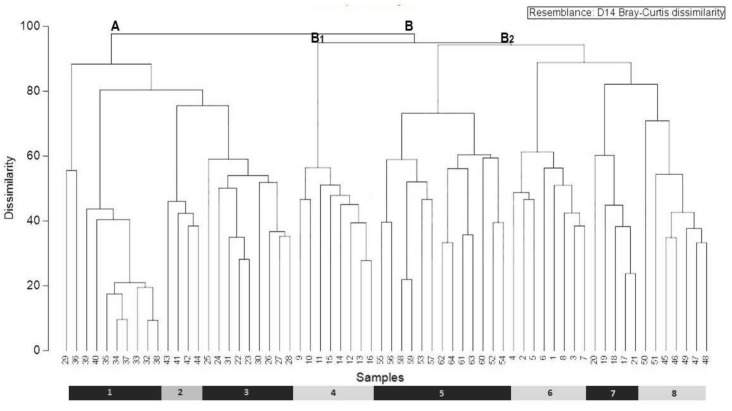
Classification analysis (UPGMA clustering dendrogram (with Bray–Curtis distance) on chasmo-chomophytic associations from the Southwestern Iberian Peninsula (included in both *Rumici indurati-Dianthion lusitani* and *Calendulo lusitanicae-Antirrhinion linkiani* alliances): 1—*Silene acutifoliae-Dianthetum lusitani* (32–40); 2—*Sileno montistellensis-Dianthetum lusitani* (41–44); 3—*Digitali thapsi-Dianthetum lusitani* (22–31); 4—*Antirrhinetum onubensis* (9–16); 5—*Sileno longiciliae-Antirrhinetum linkiani* (52–64); 6—*Sanguisorbo rupicolae-Dianthetum crassipedis* (1–8); 7—*Phagnalo saxatilis-Rumicetum indurati* (17–21); 8—*Phagnalo saxatilis-Dianthetum barbati* (45–51).

**Figure 3 plants-10-01590-f003:**
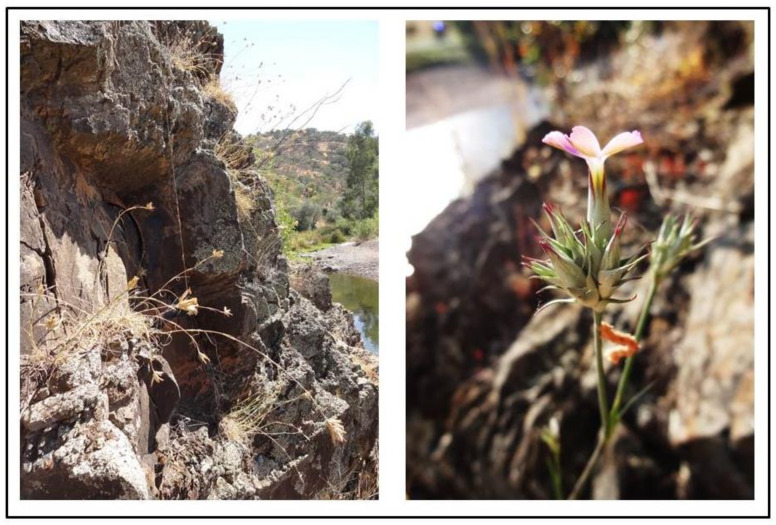
*Sanguisorbo rupicolae-Dianthetum crassipedis* on rocky cliff fissures in the lower Guadiana basin (Ribeira da Foupanilha, near Vaqueiros—Alcoutim), included in the southwestern part of the Mariánica Range Sector.

**Figure 4 plants-10-01590-f004:**
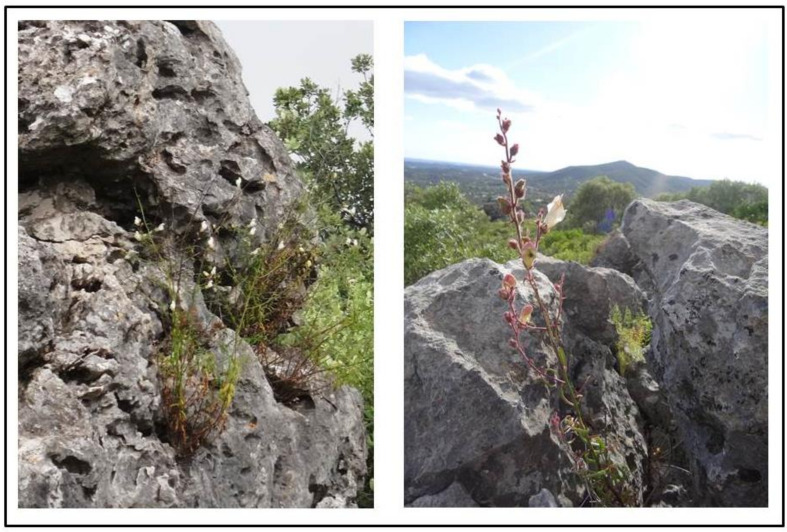
*Antirrhinetum onubensis* on rocky fissures of limestones in the Algarve and Monchique Sector—Barrocal algarvio (left: Varejota, Parragil—Loulé; right: Cerro da Cabeça, Moncarapacho—Olhão).

**Table 1 plants-10-01590-t001:** Synoptic table of southwestern Iberian Peninsula chasmo-chomophytic communities linked to the *Rumici indurati-Dianthion lusitani* and *Calendulo lusitanicae-Antirrhinion linkiani* alliances.

Association No.	1	2	3	4	5	6	7	8
**Association Characteristics**								
*Dianthus crassipes* R. Roem.	V	.	.	.	.	.	.	.
*Sanguisorba rupicola* (Boiss. & Reut.) A. Braun & C.D. Bouché	IV	II	.	.	.	.	II	.
*Rumex induratus* Boiss. & Reut.	IV	.	V	IV	.	.	.	.
*Phagnalon saxatile* (L.) Cass.	IV	III	V	.	.	.	V	III
*Antirrhinum onubensis* (Fern. Casas) Valdés	.	V	.	.	.	.	.	.
*Sedum mucizonia* (Ortega) Raym.-Hamet	.	IV	.	.	.	.	.	.
*Dianthus lusitanus* Brot.	.	.	III	V	V	V	.	.
*Anarrhinum bellidifolium* (L.) Willd.	.	.	.	III	.	.	.	.
*Digitalis thapsi* L.	.	.	.	V	II	.	.	.
*Silene acutifolia* Link ex Rohrb.	.	.	.	.	V	.	.	.
*Conopodium majus* subsp*. marizianum* (Samp.) López Udías & Mateo	.	.	.	.	V	.	.	.
*Silene x montistellensis* M. Ladero et al.	.	.	.	.	.	V	.	.
*Narcissus rupicola* Dufour	.	.	.	.	.	V	.	.
*Dianthus barbatus* L.	.	.	.	.	.	.	V	.
*Antirrhinum linkianum* Boiss. & Reuter	.	.	.	.	.	.	V	V
*Calendula suffruticosa* subsp*. Lusitanica* (Boiss.) Ohle	.	.	.	.	.	.	.	IV
*Avenula lodunensis* subsp*. occidentalis* (Delastre) Kerguélen	.	.	.	.	.	.	.	III
*Arabis sadina* (Samp.) Cout.	.	.	.	.	.	.	.	II
*Rumex intermedius* DC.	.	.	.	.	.	.	.	II
*Saxifraga cintrana* Willk.	.	.	.	.	.	.	.	I
**Alliance, Order and Class Characteristics**								
*Erysimum linifolium* Rivas Goday & Bellot	.	.	III	.	.	.	.	.
*Antirrhinum graniticum* Rothm.	.	.	III	I	.	.	.	.
*Sedum hirsutum* All.	.	.	III	.	V	III	.	.
**Companions**								
*Lavandula luisieri* (Rozeira) Rivas Mart.	IV	.	.	.	.	.	.	.
*Genista polyanthos* R. Roem. Ex Willk.	IV	.	.	.	.	.	.	.
*Cistus ladanifer* L.	IV	.	.	.	.	.	.	.
*Rosmarinus officinalis* L.	III	.	.	.	.	.	.	.
*Scilla autumnalis* L.	III	.	.	.	.	.	.	.
*Phlomis purpurea* L.	III	.	.	.	.	.	.	.
*Cheilanthes maderensis* Lowe	III	.	.	.	.	.	.	.
*Quercus rotundifolia* Lam.	II	.	.	.	.	.	.	.
*Polypodium interjectum* Shivas	II	.	.	.	.	.	.	.
*Leucojum autumnale* L.	II	.	.	.	.	.	.	.
*Campanula lusitanica* L.	II	.	.	.	.	.	.	.
*Cheilanthes guanchica* Bolle	II	.	.	.	.	.	.	.
*Pistacia lentiscus* L.	II	.	.	.	.	.	.	.
*Olea europaea* var*.sylvestris* (Mill.) Rouy ex Hegi	II	.	.	.	.	.	.	.
*Cosentinia vellea* (Aiton) Todaro	II	.	.	.	.	.	.	.
*Centaurea melitensis* L.	II	.	.	.	.	.	.	.
*Thymus mastichina* (L.) L.	III	II	.	.	.	.	.	.
*Dactylis hispanica* subsp*. lusitanica* (Stebbins & Zohary) Rivas Mart. & Izco	III	.	.	.	.	.	III	.
*Sedum forsterianum* Sm.	III	.	.	.	.	.	.	I
*Rhamnus oleoides* L.	II	III	.	.	.	.	.	.
*Hyparrhenia sinaica* (Delile) Llauradó ex G. López	II	.	.	.	.	.	III	.
*Helichrysum stoechas* (L.) Moench	II	.	.	.	.	.	III	.
*Rhamnus alaternus* L.	II	.	.	.	.	.	.	I
*Umbilicus rupestris* (Salisb.) Dandy	III	II	IV	IV	V	III	.	.
*Prasium majus* L.	.	IV	.	.	.	.	.	.
*Asplenium petrarchae* (Guérin) DC.	.	IV	.	.	.	.	.	.
*Genista hirsuta* subsp*. algarbiensis* (Brot.) Rivas Mart. et al.	.	III	.	.	.	.	.	.
*Aristolochia baetica* L.	.	III	.	.	.	.	.	.
*Asparagus albus* L.	.	III	.	.	.	.	.	.
*Juniperus turbinata* Guss.	.	II	.	.	.	.	.	.
*Valantia hispida* L.	.	II	.	.	.	.	.	.
*Theligonum cynocrambe* L.	.	II	.	.	.	.	.	.
*Pistacia terebinthus* L.	.	II	.	.	.	.	.	.
*Elaeoselinum tenuifolium* (Lag.) Lange in Willk. & Lange	.	II	.	.	.	.	.	.
*Ceratonia siliqua* L.	.	II	.	.	.	.	.	.
*Campanula erinus* L.	.	II	.	.	.	.	.	I
*Sedum sediforme* Jacq.) Pau	.	IV	.	.	.	.	.	II
*Asplenium billotii* F.W. Schultz	.	.	IV	.	I	.	.	.
*Melica minuta* L.	.	IV	.	.	.	.	III	III
*Polypodium cambricum* L.	.	III	.	.	I	.	.	I
*Galium glaucum* subsp*. australe* L.	.	.	V	.	.	.	.	.
*Cheilanthes hispanica* Mett.	.	.	II	.	.	.	.	.
*Sanguisorba minor* Scop.	.	.	II	.	.	.	.	.
*Lavandula pedunculata* (Mill.) Cav.	.	.	II	.	.	.	.	.
*Isatis platyloba* Link ex Steud.	.	.	II	.	.	.	.	.
*Lactuca serriola* L.	.	.	II	.	.	.	.	.
*Linaria saxatilis* (L.) Chaz.	.	.	II	.	.	.	.	.
*Sedum album* L.	.	.	II	.	.	.	III	III
*Sesamoides purpurascens* (L.) G. López	.	.	.	IV	.	.	.	.
*Linaria nivea* Boiss. & Reuter	.	.	.	III	.	.	.	.
*Arrhenatherum elatius* subsp*. elatius* (L.) P. Beauv. ex J. Presl & C. Presl	.	.	.	III	.	.	.	.
*Allium scorzonerifolium* Desf.	.	.	.	II	.	.	.	.
*Bufonia macropetala* Willk.	.	.	.	II	.	.	.	.
*Hieracium castellanum* Boiss. & Reut.	.	.	.	II	.	.	.	.
*Leucanthemopsis pallida* (Mill.) Heywood	.	.	.	I	.	.	.	.
*Asplenium septentrionale* (L.) Hoffm.	.	.	.	I	.	.	.	.
*Agrostis truncatula* Parl.	.	.	.	I	.	.	.	.
*Koeleria crassipes* Lange	.	.	.	I	.	.	.	.
*Jasione sessiliflora* Boiss. & Reut.	.	.	.	III	.	IV	.	.
*Biscutella valentina* (Loefl. ex L.) Heywood	.	.	.	III	.	.	.	IV
*Lactuca viminea* subsp*. chondrilliflora* Boreau	.	.	.	II	.	.	.	II
*Scrophularia canina* L.	.	.	.	II	.	.	II	.
*Polypodium vulgare* L.	.	.	.	I	.	III	.	.
*Sedum brevifolium* DC.	.	.	.	I	V	III	.	.
*Arrhenatherum elatius* subsp*. sardoum* (Em.Schmid) Gamisans	.	.	.	.	V	.	.	.
*Hypochaeris radicata* L.	.	.	.	.	V	.	.	.
*Armeria x francoi* J.C. Costa Capelo	.	.	.	.	IV	.	.	.
*Rumex acetosella* subsp*. angiocarpus* (Murb.) Murb.	.	.	.	.	II	.	.	.
*Pteridium aquilinum* L.	.	.	.	.	II	.	.	.
*Quercus pyrenaica* Willd.	.	.	.	.	I	.	.	.
*Calluna vulgaris* (L.) Huds.	.	.	.	.	I	.	.	.
*Micropyrum tenellum* (L.) Link.	.	.	.	.	I	.	.	.
*Festuca indigesta* Lag. x Lange	.	.	.	.	.	V	.	.
*Digitalis purpurea* L.	.	.	.	.	.	IV	.	.
*Juniperus communis* subsp*. alpina* (Suter) Celakn	.	.	.	.	.	IV	.	.
*Agrostis delicatula* Pourr. ex Lapeyr.	.	.	.	.	.	III	.	.
*Murbeckiella boryi* (Boiss.) Rothm.	.	.	.	.	.	III	.	.
*Phalacrocarpum oppositifolium* (Brot.) Willk.	.	.	.	.	.	III	.	.
*Micromeria juliana* (L.) Rchb.	.	.	.	.	.	.	IV	.
*Urginea maritima* (L.) Baker	.	.	.	.	.	.	III	.
*Carex hallerana* Asso	.	.	.	.	.	.	II	.
*Thymus sylvestris* Hoffmans. & Link	.	.	.	.	.	.	II	.
*Linaria supina* (L.) Mill.	.	.	.	.	.	.	III	I
*Piptatherum miliaceum* (L.) Coss.	.	.	.	.	.	.	III	I
*Melica magnolii* (Gren. & Godr.) K. Richt.	.	.	.	.	.	.	II	III
*Sanguisorba verrucosa* (Link ex G. Don) Ces	.	.	.	.	.	.	.	IV
*Galium lucidum* subsp*. fruticescens* All.	.	.	.	.	.	.	.	IV
*Silene gracilis* DC.	.	.	.	.	.	.	.	III
*Ceterach officinarum* DC.	.	.	.	.	.	.	.	III
*Geranium purpureum* Vill.	.	.	.	.	.	.	.	III
*Bituminaria bituminosa* (L.) C.H. Stirt	.	.	.	.	.	.	.	II
*Coincya johnstonii* (Samp.) Greuter & Burdet	.	.	.	.	.	.	.	II
*Scabiosa columbaria* subsp*. affinis* (Gren. & Godr.) Nyman	.	.	.	.	.	.	.	II
*Crambe hispanica* L.	.	.	.	.	.	.	.	II
*Lagurus ovatus* L.	.	.	.	.	.	.	.	II
*Anthyllis vulneraria* subsp*. maura* (G. Beck) Lindb.	.	.	.	.	.	.	.	II
*Hyparrhenia hirta* (L.) Stapf in Prain	.	.	.	.	.	.	.	II
*Allium paniculatum* L.	.	.	.	.	.	.	.	II
*Asplenium trichomanes* subsp*. quadrivalens* D.E. Mey	.	.	.	.	.	.	.	I
*Asplenium ruta-muraria* L.	.	.	.	.	.	.	.	I
*Selaginella denticulata* (L.) Link	.	.	.	.	.	.	.	I
*Hedera maderensis* subsp*. iberica* Mc Allister	.	.	.	.	.	.	.	I
*Brachypodium distachyon* (L.) Beauv.	.	.	.	.	.	.	.	I
*Blackstonia perfoliata* (L.) Huds.	.	.	.	.	.	.	.	I
*Avena barbata* Pott ex Link	.	.	.	.	.	.	.	I
*Crepis taraxacifolia* Thuill.	.	.	.	.	.	.	.	I
*Arrhenatherum album* (Vahl) W.D. Clayton	.	.	.	.	.	.	.	I
*Torilis nodosa* (L.) Gaertn.	.	.	.	.	.	.	.	I
*Sideritis hirsuta* L.	.	.	.	.	.	.	.	I
*Hyacinthoides hispanica* (Mill.) Rothm.	.	.	.	.	.	.	.	I
*Cheirolophus sempervirens* (L.) Pomel	.	.	.	.	.	.	.	I
*Iberis microcarpa* (Franco & P. Silva) Rivas Mart.	.	.	.	.	.	.	.	I
*Clinopodium vulgare* L.	.	.	.	.	.	.	.	I

**Association:** No. 1 *Sanguisorbo rupicolae-Dianthetum crassipedis* (Table 2 of this paper, 8 relevés; clusters 1–8); No. 2 *Antirrhinetum onubensis* (Table 3 of this paper, 8 relevés; 9–16); No. 3 *Phagnalo saxatilis-Rumicetum indurati* Rivas-Martínez ex F. Navarro & C. Valle in Ruiz 1986 ([20]: Table 12, 5 relevés; clusters 17–21); No. 4 *Digitali thapsi-Dianthetum lusitani* Rivas-Martínez ex V. Fuente 1986 ([21]: Table 12, 10 relevés; clusters 22–31); No. 5 *Silene acutifoliae-Dianthetum lusitani* Vicente & Galán 2008 ([22]: Table 3, 9 relevés; clusters 32–40); No. 6 *Sileno montistellensis-Dianthetum lusitani* Rivas-Martínez 1981 corr. Ladero, Rivas-Martínez, Amor, M.T. Santos & Alonso 1999 ([23]: Table 6, 4 relevés; clusters 41–44); No. 7 *Phagnalo saxatilis-Dianthetum barbati* C. Lopes, Pinto-Gomes, Lousã & Ladero 2012 ([24]: Table 20, 6 relevés; clusters 45–51); No. 8 *Sileno longiciliae-Antirrhinetum linkiani* Ladero, C. Valle, M.T. Santos, Amor, Espírito Santo, Lousã & J.C. Costa 1991 ([25]: Table 1, 13 relevés clusters 52–64). The highlighted cells indicate the characteristic species of each association.

**Table 2 plants-10-01590-t002:** Sanguisorbo rupicolae-Dianthetum crassipedis ass. nova hoc. loco (Rumici indurati-Dianthion lusitani, Phagnalo saxatilis-Rumicetalia indurati and Phagnalo saxatilis-Rumicetea indurati).

**Relevé No.**	**1**	**2**	**3**	**4**	**5**	**6**	**7**	**8 ***	**PRESENCES**
**Surface (m^2^)**	**20**	**30**	**25**	**10**	**25**	**35**	**25**	**30**	
**Altitude (m)**	**160**	**210**	**210**	**270**	**185**	**135**	**215**	**220**	
**Cover Rate (%)**	**70**	**50**	**70**	**50**	**35**	**35**	**55**	**70**	
**Orientation**	**NE**	**E**	**O**	**O**	**NE**	**O**	**NO**	**O**	
**Slope (%)**	**35**	**60**	**80**	**50**	**25**	**65**	**35**	**80**	
**Average Height (m)**	**0.4**	**0.3**	**0.4**	**0.35**	**0.4**	**0.3**	**0.4**	**0.4**	
**No. of Species**	**11**	**6**	**9**	**10**	**6**	**8**	**8**	**12**	
**Association Characteristics of Association and Higher Units**									
*Dianthus crassipes*	4	3	4	3	2	2	3	**4**	**V**
*Sanguisorba rupicola*	+	-	-	1	-	+	+	**+**	**IV**
**Alliance and Higher Ranks Characteristics**									
*Phagnalon saxatile*	-	+	+	1	+	-	-	**+**	**IV**
*Rumex induratus*	-	-	-	-	+	+	+	**+**	**III**
**Companions**									
*Lavandula luisieri*	-	1	-	+	+	+	-	**r**	**IV**
*Genista polyanthos*	r	-	r	-	-	+	r	**r**	**IV**
*Cistus ladanifer*	+	+	+	-	-	-	+	**-**	**III**
*Rosmarinus officinalis*	-	-	+	-	-	+	-	**+**	**II**
*Dactylis hispanica* subsp. *lusitanica*	+	-	-	+	+	-	-	**-**	**II**
*Umbilicus rupestris*	-	-	+	-	-	-	+	**+**	**II**
*Scilla autumnalis*	1	-	+	-	-	-	+	**-**	**II**
*Phlomis purpurea*	-	-	+	+	-	-	-	**-**	**II**
*Thymus mastichina*	-	+	-	1	-	-	-	**-**	**II**
*Sedum forsterianum*	1	-	1	-	-	-	-	**-**	**II**
*Cheilanthes maderensis*	-	-	-	+	-	-	r	**-**	**II**

**Other taxa**—Companions: +*Quercus rotundifolia*, +*Rhamnus oleoides*, +*Polypodium interjectum*, +*Leucojum autumnale* in 1; +*Campanula lusitanica* in 2; 1 *Cheilanthes guanchica*, r *Rhamnus alaternus* in 4; +*Helichrysum stoechas* in 5; +*Pistacia lentiscus*, +*Olea europaea* var. *sylvestris* in 6; +*Hyparrhenia sinaica*, +*Cosentinia vellea*, +*Asparagus albus*, +*Centaurea melitensis* in 8. **Location of the relevés**: 1—Monte da Ribeira (near Cachopo; lat 37°17′45.23″ N, long 7°44′54.44″ W); 2—Pão Duro (near Vaqueiros; lat 37°23′08.67″ N, long 7°44′54.94″ W); 3—Madeiras (lat 37°20′02.44″ N, long 7°43′38.15″ W); 4—Tavilhão (near Ameixial; lat 37°22′31.81″ N, long 8°00′00.15″ W); 5—Azinhosa (near Relvais; lat 37°18′14.40″ N, long 7°44′24.49″ W); 6—Galego (lat 37°20′17.16″ N, long 7°43′26.65″ W; 7—Plenganas (near Vaqueiros; lat 37°24′15.03″ N, long 7°44′16.45″ W); 8 (* holotypus)—Madeiras (lat 37°20′06.03″ N, long 7°43′38.82″ W).

**Table 3 plants-10-01590-t003:** Antirrhinetum onubensis ass. nova hoc. loco (Calendulo lusitanicae-Antirrhinion linkiani, Phagnalo saxatilis-Rumicetalia indurati and Phagnalo saxatilis-Rumicetea indurati).

**Relevé No.**	**1**	**2**	**3**	**4**	**5**	**6 ***	**7**	**8**	**PRESENCES**
**Surface (m^2^)**	**15**	**10**	**20**	**15**	**10**	**20**	**15**	**15**	
**Altitude (m)**	**235**	**245**	**380**	**250**	**240**	**245**	**260**	**250**	
**Cover rate (%)**	**30**	**20**	**20**	**30**	**35**	**30**	**35**	**30**	
**Orientation**	**NO**	**S**	**S**	**N**	**SO**	**N**	**O**	**S**	
**Slope (%)**	**70**	**80**	**30**	**40**	**60**	**80**	**50**	**60**	
**Average Height (m)**	**0.5**	**0.4**	**0.4**	**0.5**	**0.5**	**0.5**	**0.4**	**0,5**	
**No. of Species**	**8**	**6**	**9**	**7**	**5**	**10**	**6**	**9**	
**Association Characteristic**									
*Antirrhinum onubensis*	3	2	2	3	3	3	3	3	**V**
**Alliance and Higher Rank Characteristics**									
*Sedum mucizonia*	-	-	+	1	1	+	+	1	**IV**
*Phagnalon saxatile*	-	-	-	-	-	r	+	+	**II**
*Sanguisorba rupicola*	-	+	-	-	-	-	-		**I**
**Companions**									
*Melica minuta*	+	+	+	-	-	+	-	+	**IV**
*Asplenium petrarchae*	-	+	-	+	r	+	-	+	**IV**
*Sedum sediforme*	-	-	+	+	1	-	-	+	**III**
*Prasium majus*	+	+	+	+	-	-	-	-	**III**
*Genista hirsuta* subsp. *algarbiensis*	+	-	-	-	+	+	-	+	**III**
*Asplenium ceterach*	+	-	-	-	-	+	+	-	**II**
*Aristolochia baetica*	-	-	+	+	-	+	-	-	**II**
*Rhamnus oleoides*	+	-	-	-	-	1	-	-	**II**
*Asparagus albus*	-	-	+	-	-	+	-	-	**II**
*Polypodium cambricum*	-	-	-	-	-	+	+	-	**II**

**Other taxa**—Companions: 1 *Juniperus turbinata*, +*Valantia hispida* in 1; +*Theligonum cynocrambe* in 2; +*Campanula erinus*, +*Umbilicus rupestris* in 3; +*Thymus mastichina* in 4; +*Pistacia terebinthus* in 7; +*Elaeoselinum tenuifolium*; +*Ceratonia siliqua* in 8. **Location of the relevés**: 1—Barrocal da Tôr (near Tôr; lat 37°10′24.9″ N, long 8°02′26.9″ W) ; 2—Cerro da Cabeça (near Moncarapacho; lat 37°06′33.9″ N, long 7°46′54.8″ W); 3—Rocha da Pena (near Salir); 4—Malhada Velha (near Loulé; lat 37°10′20.4″ N, long 8°00′37.0″ W); 5—Varejota (near Parragil; lat 37°10′19.3″ N, long 8°04′50.3″ W); 6 (*holotypus)—Barrocal da Tôr (near Tôr; lat 37°10′25.3″ N, long 8°02′39.4″ W); 7—Malhada Velha (near Loulé; 37°10′18.7″ N 8°00′44.2″ W); 8—Varejota (near Parragil; 37°10′16.8″ N 8°04′47.6″ W).

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
