# Peer review of "Contribution to the Knowledge of Rocky Plant Communities of the Southwest Iberian Peninsula"

_plants, 2021, doi:10.3390/plants10081590_

Round 1

Reviewer 1 Report

The first statement of the Introduction is somehow odd and deviates from the topic. Surely habitats should be studied, but the point of the research are not habitats but plant communities – they are on target. You should focus on them or put the habitats forward, but the chasmophytic or just rocky.

It needs to be explained what is chasmo-chomophytic communities as the wide audience is not familiar with that term.

In my opinion there should be included a description of study area with detailed characterisation of the habitats, bioclimate and vegetation in the mountain ranges.

In M&M, the table with the Braun-Blanquet codes is not indispensable. But, the text should be supplemented with the averages of the cover codes that were used in computations.

The International Code for Phytosociological Nomenclature should be cited in the section devoted to nomenclature. What is the reference for the species nomenclature.

What about mosses? Bryophytes are often important contributors of the chasmophytic communities, they should be also included in samples.

Results. I must admit that 64 releves including 135 species is not many for analysing the region and establishing two new association, although it may be acceptable. Also the number of releves within the clusters that are designed to be new association is scares.

In my opinion the releve area is much to large. 20-30 square meter is a good surface for grasslands or even shrubs. For chasmophytic vegetation 1,2 or max 4 m is suitable regarding the tine microhabitat structure of this kind of habitats (fissures, clefts, ledges etc.). You need to explain why You used that large plots. Also You need to include in the Tables coordinates of the sampling plots or at least the nearest village. Otherwise the new associations are invalid as suggested by the Code.

There is a painful lack of comparisons of the obtained vegetation types in terms of species richness, diversity, pH of the habitat, microrelief of the outcrop, altitudinal amplitude and many other crucial habitat characteristic. It should be amended.

Additionally, the discussion is very scares and not informative. You should cite much more works from the Iberian Peninsula to assure the reader that You thoroughly analyse the literature.

In my opinion the description of the new associations should be based on some numerical analysis with the use of fidelity measures and assignation of the phi coefficient. Then we will be all sure that You define this association properly and the hierarchical system will be stable and firm.

The work should include the whole analytical table of all releves. It would be also useful to present at least new association on photographs.

Conclusion

Although the title of the article sounds encouraging, due to the fact that there is little data on chasmophytic vegetation from the area of ​​the SW Iberian Peninsula, further reading of the text does not allow to accept it in its present form. It needs in my opinion major revision.

Author Response

Responses to the comments and suggestions

We are grateful to the very helpful comments and suggestions provided by the reviewers for the manuscript. Several parts of the manuscript have been rewritten and the changes were made accordingly from point to point.

  1. Reviewer#1 comment:

It needs to be explained what is chasmo-chomophytic communities as the wide audience is not familiar with that term.

Authors answer

Thank you very much for suggesting. We added the necessary information in the introduction.

  1. Reviewer#1 comment:

In my opinion there should be included a description of study area with detailed characterisation of the habitats, bioclimate and vegetation in the mountain ranges.

Authors answer

Yes, you are right. We introduced a description of the study area.

  1. Reviewer#1 comment:

In M&M, the table with the Braun-Blanquet codes is (not) indispensable. But, the text should be supplemented with the averages of the cover codes that were used in computations. The International Code for Phytosociological Nomenclature should be cited in the section devoted to nomenclature.

Authors answer

Accordingly, the Table 1 has been deleted. We also incorporate the reason for applying the Van der Mareel transformation, as well as compliance with the International Code of Phytosociological Nomenclature.

  1. Reviewer#1 comment:

What about mosses? Bryophytes are often important contributors of the chasmophytic communities, they should be also included in samples.

Authors answer

Thank you very much for suggesting. Therefore, the absence of this information does not compromise the objectives of the study.

  1. Reviewer#1 comment:

In my opinion the releve area is much to large. 20-30 square meter is a good surface for grasslands or even shrubs. For chasmophytic vegetation 1,2 or max 4 m is suitable regarding the tine microhabitat structure of this kind of habitats (fissures, clefts, ledges etc.). You need to explain why You used that large plots. Also You need to include in the Tables coordinates of the sampling plots or at least the nearest village. Otherwise the new associations are invalid as suggested by the Code.

Authors answer

Yes, you are right. We introduced the coordinates of the sampling plots, as well as the nearest village and council. The area used in our study, is the real area of the plot.

  1. Reviewer#1 comment:

There is a painful lack of comparisons of the obtained vegetation types in terms of species richness, diversity, pH of the habitat, microrelief of the outcrop, altitudinal amplitude and many other crucial habitat characteristic. It should be amended.

Additionally, the discussion is very scares and not informative. You should cite much more works from the Iberian Peninsula to assure the reader that You thoroughly analyse the literature.

Authors answer

Yes, you are right. As mentioned above we introduced the information in the subsection study area, to clarify the differences between plant. We cited the most important studies for the description of communities.

  1. Reviewer#1 comment:

In my opinion the description of the new associations should be based on some numerical analysis with the use of fidelity measures and assignation of the phi coefficient. Then we will be all sure that You define this association properly and the hierarchical system will be stable and firm.

The work should include the whole analytical table of all releves. It would be also useful to present at least new association on photographs.

Authors answer

For purpose, and considering the other reviewer - the data presentation is ok – however, as requested, we discussed better the results in the Discussion. Additionally, we present photographs of the new associations.

Thank you very much for your review. We introduced many improvements, so we hope that the current version is much better to be published in Plants.

Reviewer 2 Report

The manuscript deals with chasmophytic vegetation in the Iberian Peninsula. The topic is interesting. The English requires some minor correction. The data presentation is ok, but the results should be better discussed. For example, Cluster A in the Dendrogram (Figure 2) includes 3 associations of Rumici indurati-Dianthion lusitani, whereas the other 2 associations of this alliance cluster together with the 3 associations of alliance Calendulo lusitanicae-Antirrhinion linkiani. This is noteworthy. The authors should discuss it in the Discussion. Is the alliance Rumici indurati-Dianthion lusitani with these 5 associations correctly distinguished from Calendulo lusitanicae-Antirrhinion linkiani? Is the description of a new alliance necessary? Is the Rumici indurati-Dianthion lusitani limited to 3 associations, while the other 2 associations may be included in the alliance Calendulo lusitanicae-Antirrhinion linkiani? In addition, in all the Tables it is necessary to divide the characteristic species in “Characteristic of association” and “Characteristic of alliance and higher ranks”. Therefore, I suggest reconsideration after Major revisions.

Minor points:

Line 4: Delete “7” after “Mauro Raposo 57”

Line 30 (Keywords): Do not repeat words already included in the Title, i.e. delete “Rocky plant communities”

Line 34: Add “to be”: that need to be

Line 37: Add more references: threats [3, 4, 5, 6, 7, 8, 9, 10]

  1. Wagensommer, R.P.; Fröhlich, T.; Fröhlich, M. First record of the southeast European species Cerinthe retorta Sibth. & Sm. (Boraginaceae) in Italy and considerations on its distribution and conservation status. Acta Bot. Gallica: Botany Letters 2014, 161 (2), 111–115. https://doi.org/10.1080/12538078.2014.892438.
  2. Wagensommer, R.P.; Medagli, P.; Turco, A.; Perrino, E.V. IUCN Red List evaluation of the Orchidaceae endemic to Apulia (Italy) and considerations on the application of the IUCN protocol to rare species. Nature Conservation Research 2020, 5 (suppl. 1), 90–101. https://doi.org/10.24189/ncr.2020.033.

Line 49: write the reference number instead of the year “(2012)”

Line 50: “developing” instead of “developed”

Line 57: “indurati” instead of “indurate”

Line 65: Delete one of the two “the the”

Table 1: 1/20 = 5%. Thus: 1: (from 1% to 5%); 2: (from 5% to 25%). If not: 2: Very abundant individuals or covering at least 1/10 of the surface (from 10% to 25%).

Line 75: Maybe better “Followed Nomenclature” or simply “Nomenclature”

Line 77: “Plant identification” instead of “Botanic identification”

Lines 86-87: Tables 3 and 4, not Tables 4 and 5

Line 87: “Figure 2 clusters 1-8” instead of “clusters 1-8”

Line 87: “Figure 2 clusters 9-16” instead of “clusters 9-16”

Line 97: “dendrogram” instead of “dendogram”

Figure 2: Add a symbol to identify all the 8 associations (or simply C1, C2, C3, and so on)

Figure 2: Cluster A includes 3 associations of Rumici indurati-Dianthion lusitani, whereas the other 2 associations of this alliance cluster together with the 3 associations of alliance Calendulo lusitanicae-Antirrhinion linkiani. This is noteworthy. The authors should discuss it in the Discussion. Is the alliance Rumici indurati-Dianthion lusitani with these 5 associations correctly distinguished from Calendulo lusitanicae-Antirrhinion linkiani? Is the description of a new alliance necessary? Is the Rumici indurati-Dianthion lusitani limited to 3 associations, while the other 2 associations may be included in the alliance Calendulo lusitanicae-Antirrhinion linkiani?

Line 117: “which occurs” instead of “which occur”

Line 120: “occurs mostly” instead of “occur mostly”

Table 2: Divide the characteristic species in “Characteristic of association” and adding a section “Characteristic of alliance and higher ranks”

Table 2: Do not insert species characteristic of higher ranks (e.g. Cosentinia vellea, Cheilanthes maderensis, Cheilanthes guanchica, etc.) in the Companions. Create a section “Characteristic of order and class” or simply “Characteristic of alliance and higher ranks”

Table 2: Delete the blank line after Biscutella valentina

Line 161: “Figure 2 clusters 1-8” instead of “clusters 1-8”

Line 163: “appear clearly defined in group B2” instead of “appear clearly defined in group A”

Lines 182-183: Caption to Table 3: All the names in italics

Table 3: What is the meaning of the backets ()? You do not indicate them in Table 1. What value after the transformation proposed by Van der Maarel?

Table 3: Divide the characteristic species in “Characteristic of association” and adding a section “Characteristic of alliance and higher ranks”

Line 190: “Figure 2 clusters 9-16” instead of “clusters 9-16”

Line 190: Delete the comma at the end of the line

Lines 199-200: Caption to Table 4: All the names in italics

Table 4: Relevé 6: number of species “11” instead of “10”

Table 4: What is the meaning of the backets ()? What value after the transformation proposed by Van der Maarel?

Table 4: Divide the characteristic species in “Characteristic of association” and adding a section “Characteristic of alliance and higher ranks”

Line 206: “characterized” instead of “accompanied”

Line 229: “Dianthetum lusitani” in italics

Line 250 (References): Check the journal guidelines. For example, the year should always be in bold (correct it e.g. in lines 276, 280, etc.), the Journal name should be in italics (correct it e.g. in line 276 Glob. Geobot., line 280 Itinera Geobotánica, etc.)

Line 266: “Fagonia cretica” in italics

Lines 272-273: Acer opalus / granatensis / Corylus avellana in italics

Line 317: “Phagnalo saxatilis-Dianthetum barbati” in italics

Line 324: The “F” of Fuente is written to big and in bold

Author Response

Responses to the comments and suggestions

We are grateful to the very helpful comments and suggestions provided by the reviewers for the manuscript. Several parts of the manuscript have been rewritten and the changes were made accordingly from point to point.

  1. Reviewer#2 comment:

The manuscript deals with chasmophytic vegetation in the Iberian Peninsula. The topic is interesting. The English requires some minor correction. The data presentation is ok, but the results should be better discussed. For example, Cluster A in the Dendrogram (Figure 2) includes 3 associations of Rumici indurati-Dianthion lusitani, whereas the other 2 associations of this alliance cluster together with the 3 associations of alliance Calendulo lusitanicae-Antirrhinion linkiani. This is noteworthy. The authors should discuss it in the Discussion. Is the alliance Rumici indurati-Dianthion lusitani with these 5 associations correctly distinguished from Calendulo lusitanicae-Antirrhinion linkiani? Is the description of a new alliance necessary? Is the Rumici indurati-Dianthion lusitani limited to 3 associations, while the other 2 associations may be included in the alliance Calendulo lusitanicae-Antirrhinion linkiani? In addition, in all the Tables it is necessary to divide the characteristic species in “Characteristic of association” and “Characteristic of alliance and higher ranks”. Therefore, I suggest reconsideration after Major revisions.

Authors answer

Thank you very much for suggesting. We promote a better discussion of the results obtained in cluster analysis, to better accommodate the inaccuracies detected in the subgroup B2.

  1. Reviewer#2 comment:

In addition, in all the Tables it is necessary to divide the characteristic species in “Characteristic of association” and “Characteristic of alliance and higher ranks”

Authors answer

For purpose, all the tables have been modified accordingly.

  1. Reviewer#3 comment:

Minor points:

Line 4: Delete “7” after “Mauro Raposo 57”

Line 30 (Keywords): Do not repeat words already included in the Title, i.e. delete “Rocky plant communities”

Line 34: Add “to be”: that need to be

Line 37: Add more references: threats [3, 4, 5, 6, 7, 8, 9, 10]

  1. Wagensommer, R.P.; Fröhlich, T.; Fröhlich, M. First record of the southeast European species Cerinthe retorta Sibth. & Sm. (Boraginaceae) in Italy and considerations on its distribution and conservation status. Acta Bot. Gallica: Botany Letters 2014, 161 (2), 111–115. https://doi.org/10.1080/12538078.2014.892438.
  2. Wagensommer, R.P.; Medagli, P.; Turco, A.; Perrino, E.V. IUCN Red List evaluation of the Orchidaceae endemic to Apulia (Italy) and considerations on the application of the IUCN protocol to rare species. Nature Conservation Research 2020, 5 (suppl. 1), 90–101. https://doi.org/10.24189/ncr.2020.033.

Line 49: write the reference number instead of the year “(2012)”

Line 50: “developing” instead of “developed”

Line 57: “indurati” instead of “indurate”

Line 65: Delete one of the two “the the”

Table 1: 1/20 = 5%. Thus: 1: (from 1% to 5%); 2: (from 5% to 25%). If not: 2: Very abundant individuals or covering at least 1/10 of the surface (from 10% to 25%).

Line 75: Maybe better “Followed Nomenclature” or simply “Nomenclature”

Line 77: “Plant identification” instead of “Botanic identification”

Lines 86-87: Tables 3 and 4, not Tables 4 and 5

Line 87: “Figure 2 clusters 1-8” instead of “clusters 1-8”

Line 87: “Figure 2 clusters 9-16” instead of “clusters 9-16”

Line 97: “dendrogram” instead of “dendogram”

Figure 2: Add a symbol to identify all the 8 associations (or simply C1, C2, C3, and so on)

Figure 2: Cluster A includes 3 associations of Rumici indurati-Dianthion lusitani, whereas the other 2 associations of this alliance cluster together with the 3 associations of alliance Calendulo lusitanicae-Antirrhinion linkiani. This is noteworthy. The authors should discuss it in the Discussion. Is the alliance Rumici indurati-Dianthion lusitani with these 5 associations correctly distinguished from Calendulo lusitanicae-Antirrhinion linkiani? Is the description of a new alliance necessary? Is the Rumici indurati-Dianthion lusitani limited to 3 associations, while the other 2 associations may be included in the alliance Calendulo lusitanicae-Antirrhinion linkiani?

Line 117: “which occurs” instead of “which occur”

Line 120: “occurs mostly” instead of “occur mostly”

Table 2: Divide the characteristic species in “Characteristic of association” and adding a section “Characteristic of alliance and higher ranks”

Table 2: Do not insert species characteristic of higher ranks (e.g. Cosentinia vellea, Cheilanthes maderensis, Cheilanthes guanchica, etc.) in the Companions. Create a section “Characteristic of order and class” or simply “Characteristic of alliance and higher ranks”

Table 2: Delete the blank line after Biscutella valentina

Line 161: “Figure 2 clusters 1-8” instead of “clusters 1-8”

Line 163: “appear clearly defined in group B2” instead of “appear clearly defined in group A”

Lines 182-183: Caption to Table 3: All the names in italics

Table 3: What is the meaning of the backets ()? You do not indicate them in Table 1. What value after the transformation proposed by Van der Maarel?

Table 3: Divide the characteristic species in “Characteristic of association” and adding a section “Characteristic of alliance and higher ranks”

Line 190: “Figure 2 clusters 9-16” instead of “clusters 9-16”

Line 190: Delete the comma at the end of the line

Lines 199-200: Caption to Table 4: All the names in italics

Table 4: Relevé 6: number of species “11” instead of “10”

Table 4: What is the meaning of the backets ()? What value after the transformation proposed by Van der Maarel?

Table 4: Divide the characteristic species in “Characteristic of association” and adding a section “Characteristic of alliance and higher ranks”

Line 206: “characterized” instead of “accompanied”

Line 229: “Dianthetum lusitani” in italics

Line 250 (References): Check the journal guidelines. For example, the year should always be in bold (correct it e.g. in lines 276, 280, etc.), the Journal name should be in italics (correct it e.g. in line 276 Glob. Geobot., line 280 Itinera Geobotánica, etc.)

Line 266: “Fagonia cretica” in italics

Lines 272-273: Acer opalus / granatensis / Corylus avellana in italics

Line 317: “Phagnalo saxatilis-Dianthetum barbati” in italics

Line 324: The “F” of Fuente is written to big and in bold

Authors answer

Yes, you are right. There were mistakes in the main text. We corrected it.

Thank you very much for your review. We introduced many improvements, so we hope that the current version is much better to be published in Plants.

Reviewer 3 Report

The presented manuscript reported two new communities on the territory of the southwest Iberian Peninsula.

In general, the text is presented as a short research note rather than as a full research article. The Introduction is scrace. Material and methods section needs additional clarification. There is no discussion and conclusion. The described plant communities could be linked to the abiotic factors, pressures and habitat status.

Based on the above, considerable rewriting is recommended.

Several terminological, grammatical and technical errors outlined in the pdf file.

Author Response

Responses to the comments and suggestions

We are grateful to the very helpful comments and suggestions provided by the reviewers for the manuscript. Several parts of the manuscript have been rewritten and the changes were made accordingly from point to point.

  1. Reviewer#3 comment:

In general, the text is presented as a short research note rather than as a full research article. The Introduction is scrace. Material and methods section needs additional clarification. There is no discussion and conclusion. The described plant communities could be linked to the abiotic factors, pressures and habitat status.

Authors answer

Thank you very much for suggesting. We added many improvements in the manuscript, including the mistakes in the main text.

Thank you very much for your review. We introduced many improvements, so we hope that the current version is much better to be published in Plants.

Round 2

Reviewer 1 Report

Dear Authors,

I think the corrected version is much better than the previous one. I recommend the work for publication in Plants after some minor language changes.

Best regards

Reviewer 2 Report

The authors follow most of the suggestions of the reviewers. I suggest acceptance in present form.